# Revisiting Standard-Definition Map Based Motion Prediction in the Era of End-to-End Autonomous Driving

## Abstract

Most motion prediction models use maps as environmental context. For a long time, high-definition (HD) maps are preferred as they provide detailed lane-level information and often lead to significantly better performance compared with standard-definition (SD) maps. However, offline HD maps require extensive manual annotation, making them costly and unscalable. Online mapping-based methods still require HD map annotation to train the online mapping module, which is costly as well and may suffer from the issue of out-of-distribution map elements. In this paper, we look back to SD maps in the era of end-to-end autonomous driving and focus on narrowing the performance gap between HD and SD maps. We initially extend anchor-based and anchor-free motion prediction models in an end-to-end manner and find the performance gap narrowed with the introduction of raw image features. Furthermore, we discover the unique challenges that the coarse and misaligned SD maps bring to feature fusion of the anchor-free model and on anchor generation of the anchor-based model. Thus, we design two novel modules named Enhanced Road Observation and Pseudo Lane Expansion to address these issues. With these insights, we reduce the performance gap between HD and SD maps by 84%, making SD map based motion prediction achieve comparable performance as HD map based one.

## 1 Introduction

As a crucial component in autonomous driving, most motion prediction models forecast the future states of agents based on their historical data and environmental context. Generally, there are two types of maps to provide environmental information: high-definition (HD) maps and standard-definition (SD) maps.

HD maps offer detailed and precise lane-level road geometry information, such as lane dividers, centerlines, pedestrian crossings, and stop lines, and are widely employed in motion prediction models (Gu et al., 2021; Shi et al., 2022; Zhou et al., 2022), as shown in Fig. 1(a). However, the high resolution of HD maps comes at a significant cost. Their creation requires a fleet of vehicles equipped with high-precision sensors, such as LiDAR, along with extensive manual annotation, costing approximately 4,000–5,000 \$/km (Zhu et al., 2024). Since major cities often have thousands of kilometers of roads, constructing HD maps for a single city can incur costs in the millions of dollars. Even worse, HD maps developed by different autonomous driving companies are often proprietary and difficult to share, further increasing the challenges of obtaining HD maps. To reduce the construction cost of HD maps, online mapping models (Liao et al., 2023; Li et al., 2024b; Yuan et al., 2024) are developed. These models predict HD map elements around the vehicle using sensor data in real-time, providing map input for downstream modules. Nevertheless, training these supervised models still depends on ground-truth annotations, facing the challenge of annotation and generalization.

Before HD maps are widely adopted for autonomous driving, SD maps are already used in human driving for a long time. These maps can be generated using vehicles equipped with low-cost IMU and GPS systems, combined with minimal manual processing, resulting in significantly lower acquisition costs (Mooney et al., 2017). Provided by OpenStreetMap or Google, these maps cover thousands of

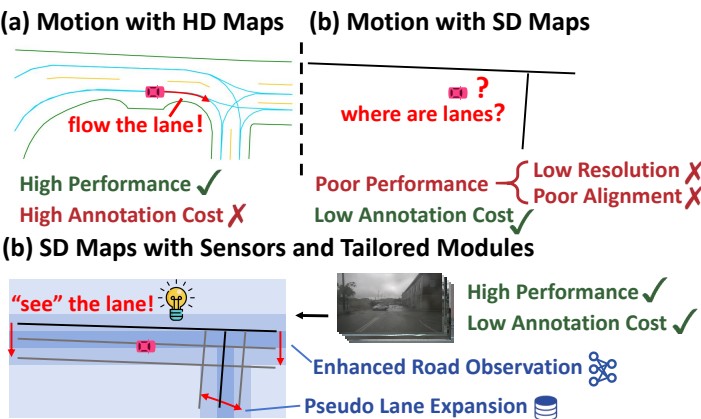

Figure 1: **Revisiting SD Maps in Motion Prediction.** (a) Lane-level HD maps provide detailed information but are expensive to build. (b) Road-level SD maps are available at low cost but lead to poor performance in previous motion prediction models due to their low resolution and poor alignment. (c) We introduce raw sensor data and design tailored modules for SD maps to reach comparable performance with HD map inputs in the same model.

cities and regions, providing information on road direction and intersection structures, and assisting in route planning and driving maneuvers like turning.

However, compared with HD maps, SD maps have two major weaknesses: (1) **Low Resolution**. SD maps only indicate the general direction of roads without providing lane-level details. A road may be represented by just one or two polylines in SD maps. (2) **Poor Alignment**. Due to localization errors, the polylines in SD maps may not align with the center of the roads and could even be outside the roads. The former issue makes SD maps provide less information and the latter makes them provide even incorrect information. Previous study (Liao et al., 2024) and our initial experiments show that substituting SD maps for HD maps in traditional non-end-to-end motion prediction models results in a significant drop in performance. This is why previous motion prediction models rarely use SD maps, as shown in Fig. 1(b).

In the era of end-to-end autonomous driving (Hu et al., 2023; Jiang et al., 2023), the performance gap between SD maps and HD maps (SD-HD gap) may narrow. End-to-end learning presents an attractive way to capture task-specific information directly from raw sensor data. These features encompass environmental information around the agent, playing a similar role as maps, thereby reducing reliance on map precision. During the process, SD maps could serve as a rough guide for feature aggregation, providing an understanding of the road's general layout, which could potentially achieve performance comparable to using HD maps with detailed road information.

Based on the motivation, we revisit SD map based motion prediction. To begin with, we formulate the problem and leverage sensor data with SD map guidance on anchor-based and anchor-free models. We extract BEV features from raw data with an encoder and incorporate several modules to efficiently fuse BEV features with agent features and SD map features. Experiments demonstrate that the SD-HD gap diminishes in the end-to-end framework.

Next, we conduct a deeper analysis of problems caused by the weaknesses of SD maps in the models on anchor-based and anchor-free models and design new modules to address them, as shown in Fig. 1(c). Anchor-based models like DenseTNT (Gu et al., 2021) select candidate goal points (anchors) from maps. The low resolution and poor alignment of SD maps result in poor distributions of goal points, for example, there may be no goal point around agents. We solve this issue by introducing an anchor generation method called **Pseudo Lane Expansion**, which generates extra pseudo anchors by groups based on original SD map instances to improve anchor distributions. For anchor-free models such as HiVT (Zhou et al., 2022) and MTR (Shi et al., 2022), due to sparse and misaligned SD maps, limited or helpless BEV features could be fused with SD map features in our original module which uses standard deformable attention. Thus, we modify the multi-head deformable attention by adding reference points and introducing a head weighting mechanism. Named **Enhanced Road Observation**, the module can sample a wider range of BEV features around SD instances.

With these insights, the proposed model narrows the SD-HD gap by 93% (minADE) and 82% (minFDE) on the anchor-based model. For the anchor-free model, the reductions are 84% and 77%.

The model also outperforms state-of-the-art end-to-end prediction model (Gu et al., 2024b) under the same protocol by 11.0% and 8.1% on minADE for anchor-based and anchor-free models.

Our contributions are threefold:

- We revisit SD map based motion prediction and improve its performance with raw sensor data, achieving results comparable to those based on HD maps and outperforming the online HD map based motion prediction models.
- We propose a BEV-SDmap interactor called Enhanced Road Observation and a goal point generation method named Pseudo Lane Expansion to improve the performance of SD map based motion prediction.
- We analyze the factors affecting the performance gap between SD and HD maps, including the type of base model and the introduction of sensor data.

## 2 RELATED WORK

### 2.1 HD MAPS AND SD MAPS

**HD Maps.** High-definition(HD) maps contain detailed road information, but their creation requires extensive manual annotation, making them expensive and unscalable (Li et al., 2022a). This leads to the development of various online map estimation methods, which estimate HD maps from camera or LiDAR data. Recent approaches such as MapTR(Liao et al., 2023; Li et al., 2024b) are mostly based on an encoder-decoder architecture, where BEV (bird's-eye view) features are extracted from sensor data, and a transformer decoder is used to predict various map elements. Online map estimation enables autonomous vehicles to operate in areas without Offline HD map coverage, reducing dependency on HD maps. However, these methods still require HD map ground truth for supervised training, making it challenging to obtain sufficient HD map data for large-scale training. Additionally, online HD map estimation consumes computational resources and time.

**SD Maps.** At present, conducting the motion prediction task with SD maps remains an underexplored research topic. A previous study (Liao et al., 2024) attempts to extract SD maps from Open Street Maps (OSM) (Haklay & Weber, 2008) and feed them to a traditional non-end-to-end motion prediction model. However, it merely substituted the map representation without adapting the motion prediction model to the characteristics of SD maps, resulting in a significant SD-HD gap. Most recent works related to SD maps treat them as priors for generating HD maps (Li et al., 2024a). For example, PriorDrive (Zeng et al., 2024) uses a unified vector encoder to effectively encode diverse vector prior maps including SD maps to enhance the robustness and accuracy of online HD map construction. Previous studies (Jiang et al., 2024; Zhang et al., 2024; Liao et al., 2024) merge SD maps from Open Street Maps (OSM) (Haklay & Weber, 2008) into widely used datasets like nuScenes (Caesar et al., 2019) and OpenLane-V2 (Wang et al., 2023) to make them more available.

### 2.2 MOTION PREDICTION

**Traditional Motion Prediction Models.** Most Traditional motion prediction models take historical data of agents and offline HD maps as inputs. Early models typically use rasterized HD maps and encode them with Convolutional Neural Networks (CNNs) (Marchetti et al., 2020; Biktairov et al., 2020; Casas et al., 2020; Gilles et al., 2021). However, high-resolution rasterized maps incur significant storage and computational costs. Recent approaches shift towards vectorized representations of HD maps. In terms of map utilization, some methods such as LaneGCN (Liang et al., 2020), GOHOME (Gilles et al., 2022) and HiVT (Zhou et al., 2022), employ Graph Neural Networks (GNNs) to encode the influence of map elements on vehicle interactions. Other methods like MTR (Shi et al., 2022) and QCNet (Zhou et al., 2023) directly use Transformer architectures, leveraging cross-attention mechanisms to fuse map and vehicle features. Some target-based approaches (Zhao et al., 2021; Gu et al., 2021) generate candidate target points based on the map, leading to a stronger dependency on the map.

**Motion Prediction with Senors.** With the advancement of end-to-end learning, many methods incorporate sensor information into the motion prediction task. Most of them focus on collaboration among various tasks in autonomous driving and adopt various input settings depending on specific problems. ViP3D (Gu et al., 2023) combines detection, tracking, and prediction in an end-to-end

Table 1: **Protocols and Focus of Works in Motion Prediction with Sensor Data.**

| Methods | Map information | Agents' Information | Focus |
|---|---|---|---|
| ViP3D (Gu et al., 2023) | GT HD maps | Detection results | The cooperative relations of detection and motion prediction |
| PiP (Jiang et al., 2022) | Mapping results | Detection results | The interaction between detection and online mapping |
| BEVPred (Gu et al., 2024b) | Mapping results | GT | Acceleration on online mapping and motion prediction |
| Ours | GT SD maps | GT | Replacing HD maps with low-cost SD maps for motion prediction |

structure, and gets agent information directly from sensor data. Its motion prediction module utilizes agent information from the detection and tracking module and GT HD maps as inputs. PiP (Jiang et al., 2022) is the first end-to-end Transformer-based framework that jointly and interactively performs online mapping, object detection, and motion prediction. Its motion prediction module obtains both types of information from the up-streaming modules, which is suitable for exploring the interaction between detection and online mapping, and its influence on the downstream motion prediction task. BEVPred (Gu et al., 2024b) takes BEV features from a pre-trained online mapping model as map information for acceleration on online mapping and motion prediction. It uses ground-truth as agents' information. However, there is a lack of end-to-end motion prediction methods specifically designed for or adapted to SD maps.

**Motion Prediction with Online HD Maps.** Directly inputting online HD maps into motion prediction models is a basic method of online mapping based motion prediction. However, the error between estimated HD maps and GT HD maps leads to errant behaviors in motion prediction. To address this issue, a highly rated work (Gu et al., 2024a) extends online map estimation methods to additionally estimate uncertainty, to provide information about potential errors of maps for downstream models. BEVPred (Gu et al., 2024b) mentioned above is also an online mapping based motion prediction model. This line of research has the same goal as SD map based motion prediction, which is to reduce the cost of maps for motion prediction, but they have different paradigms and protocols. SD map based motion prediction takes low-cost SD maps as input, while online mapping based motion prediction uses high-cost HD map annotation for training and needs extra computation for online map generation.

## 3 METHOD

### 3.1 PROBLEM FORMULATION

As discussed in Sec. 2.2 and Table 1, current end-to-end motion prediction models adopt different protocols based on their focus. Unlike studies that focus on multitask cooperation, we introduce sensor data to improve performance with SD maps. To eliminate interference from inaccurate detection, we use GT agent information and GT SD maps along with raw camera data as inputs.

### 3.2 LEVERAGING SENSOR DATA WITH SD MAP GUIDANCE

**Encoding Raw Sensor Data.** We adopt BEVFormer (Li et al., 2022b) as an encoder to extract BEV features. The encoder processes multi-view images with an image backbone (e.g., ResNet50) and then uses BEV2PV look-up to construct BEV features $B \in \mathbb{R}^{H \times W \times C}$ with the intrinsic and extrinsic of each camera. Note that other BEV encoders could work as well, for example, LSS (Philion & Fidler, 2020).

**SD Map Guided Road Observation.** The BEV feature represents the visual context of the surrounding environment spatially, while the SD map provides the coarse location of the road. We use the SD map as a guide to "observe" its vicinity, aggregating nearby BEV features along the SD map to obtain more detailed road information. Specifically, we use deformable attention (Zhu et al., 2020) for this process. We denote the vectorized SD map as $m \in \mathbb{R}^{N_m \times 2}$, with the encoded map features represented by $F_{map} \in \mathbb{R}^{N_m \times D}$, where $N_m$ is the total number of points constituting the polylines. $T_{\text{ref}}$ is the translation from the ego vehicle coordinate system to BEV grids. The BEV-SDmap feature is obtained via

$$F_{\text{BEV-SDmap}} = \text{DeformAtt}(F_{map}, T_{ref}(m), B) \quad (1)$$

The fused BEV-SDmap features are then used for the generation of local embeddings in HiVT and for anchor selection in DenseTNT, as shown in Fig. 2 (upper) and Fig. 4 (left). The agent features are fused with BEV features through similar progress.

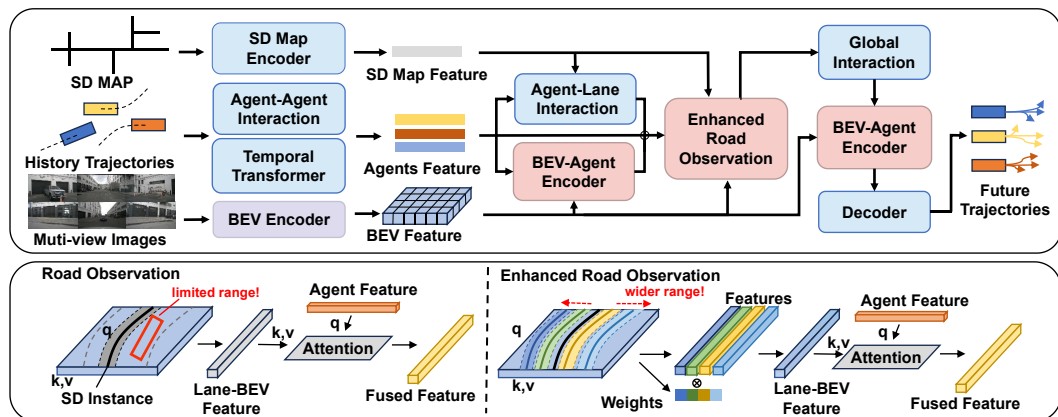

Figure 2: **Anchor-Free Model**. Upper: Overall structure. We integrate Enhanced Road Observation (improved from SD map guided road observation) and BEV-Agent encoder in the HiVT model. Lower: Enhanced Road Observation. Due to the low resolution of SD maps, the coverage of the original SD map guided road observation using standard deformable attention is limited. Enhanced Road Observation, by extending reference points along parallel lines, allows sampling of BEV features over larger area.

### 3.3 ENHANCING FEATURE FUSION WITH SD MAPS IN ANCHOR-FREE MODEL

**Challenges in Feature Fusion with SD Maps.** We further analyze the performance of SD map guided road observation in the anchor-free model HiVT. When the road is narrow and the SD lane is at the center of the road, the sample points around the reference points (generated from SD lanes) in deformable attention can almost cover the entire road. However, when the road is wide and there are only one or two SD lanes, these points are in a limited region and can not cover some key locations such as the road edge, as shown in Fig. 2 (lower left). Even worse, the sample points from misaligned SD lanes may capture useless BEV features around them, such as those far away from the road.

**Enhanced Road Observation.** We observe that SD maps accurately indicate road direction despite alignment errors. Thus, we aim to expand reference points by creating parallel lines. In this way, a single line gets a wide sample area. Even if the original line from the SD map is out of the road, some of its parallel lines may still be within the road and sample useful BEV features. Based on this insight, we propose **Enhanced Road Observation**. We denote the SD map $S \in \mathbb{R}^{N_s \times N_d \times 2}$ with $N_s$ SD map instances and each instance consist of $N_d$ points. For $S_i$, we generate the extended reference points $\hat{S} \in \mathbb{R}^{N_s \times N_l \times N_d \times 2}$ through:

$$\hat{S}_{ij} = \text{Parallel}(S_i, l_j) \tag{2}$$

For a single SD map instance $S_i$, a set of parallel lines $\{\hat{S}_{i1}, \hat{S}_{i2}, \ldots, \hat{S}_{iN_l}\}$ are generated. Where $N_l$ is the number of parallel lines and $l_j$ represents the distance between the original polyline and each parallel line. To simplify processing, the number of parallel lines $N_l$ is set equal to the number of heads $N_h$ in deformable attention.

Next, we modify deformable attention to accommodate multiple sets of reference points. In standard deformable attention, multiple heads share a single set of reference points, while **we assign each head with its own set of reference points corresponding to a parallel line**. Since not all heads extract meaningful features (e.g., some parallel lines may lie outside the road), we apply a learnable weight parameter $h_j$ to the features obtained by each head:

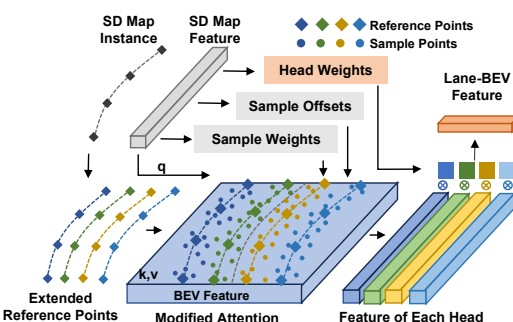

Figure 3: **Enhanced Road Observation.** We extend several sets of reference points for deformable attention and predict the weight for each set to cover lanes missed in SD maps.

$$F_{\text{BEV-SDmap}} = \sum_{j=1}^{N_h} h_i W_j \left[ \sum_{k=1}^{K} A_{i,j,k} \cdot W_j' B \left( \hat{s}_{i,j} + \Delta \hat{s}_{i,j,k} \right) \right] \tag{3}$$

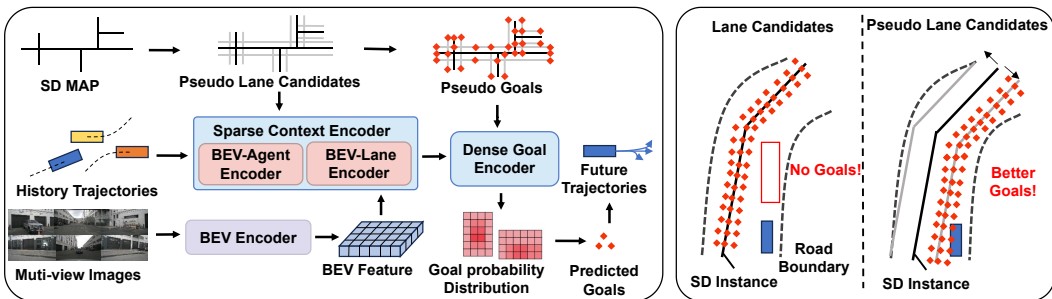

Figure 4: **Anchor-Based Model.** Left: Overall structure. We integrate Pseudo Lane Expansion and feature fusion modules in the DenseTNT model. Right: Original SD map vs. Pseudo Lane Expansion. Target points generated directly from the SD map may not be near the agent, while Pseudo Lane Expansion generates pseudo target points along the road direction near the agent.

Where $K$ denotes the number of sample points. $W$ and $W' \in \mathbb{R}^{D \times D}$ are linear projections applied to the BEV features $B$. The sample points $A \in \mathbb{R}^{N_d \times N_h \times K}$, sampling offsets $\Delta \hat{s} \in \mathbb{R}^{N_d \times N_h \times K}$, and head weights $h$ are all obtained through different linear projections from the SD map features $Q_s$, which serve as the query. As in standard deformable attention, the weights $A$ are normalized using softmax along the last dimension, shown in Fig. 3.

For the head weights $h$, multiple parallel lines from an SD instance may capture environmental information in different lanes, and softmax would overly prioritize a single head. DQNv4 (Xiong et al., 2024) discusses the need for softmax normalization in attention and claims that normalization becomes unnecessary when the degradation issue does not exist. Thus, we do not use functions such as Sigmoid of Softmax to normalize head weights.

### 3.4 IMPROVING ANCHOR DISTRIBUTION FROM SD MAPS IN ANCHOR-BASED MODEL

**Challenges in Anchor Generation with SD Maps.** In the anchor-based model DenseTNT, the challenges of using SD maps lie in anchor generation. The model densely samples points around candidate target points from a vectorized map, then predicts the probability for each, selecting the final target point based on these probabilities. As a result, the map has a direct impact on the distribution of candidate target points. Unfortunately, due to the low resolution and alignment accuracy of SD maps, there may be no candidate points near the agent or its future trajectory, which significantly reduces prediction accuracy. Fig. 4 shows the matter.

**Pseudo Lane Expansion.** To address this issue, adding extra pseudo anchors is a straightforward way. Since DenseTNT is designed around lane and goal features, we directly input the expanded SD lines into the model. We denote a single polyline in the SD map containing $N_d$ 2D points $p \in \mathbb{R}^{N_d \times 2}$, the unit normal vector of the vector from the i-th point to the (i+1)-th point $\mathbf{n_i}$. The i-th point $\hat{p}_{ji}$ of the j-th expanded line is calculated as:

$$\hat{p}_{ji} = p_i + d_j \mathbf{n_i} \tag{4}$$

where $d_j$ is the distance between the j-th expanded line and the original line. Because DenseTNT's original lane/goal scoring module predicts weights for each target point, we no longer predict weights for each parallel line as Enhanced road observation. This simple but effective anchor generation method greatly improves prediction performance with SD maps. We demonstrate the process as in Fig. 4 (right).

**Adaptive Pseudo Lane Expansion.** Because of the variability in road structures and the distribution of SD maps, using fixed parameters, including the number and distances of the extended parallel lines in Pseudo Lane Expansion, often fails to achieve optimal performance across diverse scenarios. Thus, we adapt the parameters based on the distance of the SD map lane relative to the vehicle to predict, and the density of SD map lanes. **(1) Distance.** If the closest distance between the SD map and the target vehicle is large, it likely indicates poor alignment of the SD map. In such case, we generate more pseudo lanes on the side closer to the vehicle while reducing the number on the opposite side to decrease the creation of irrelevant lines, as in Fig. 5 (left). **(2) Density.** For sparsely distributed SD lanes (only one or two SD lanes in the range), we increase the number of pseudo lanes to ensure better coverage of the road. Conversely, in dense areas (e.g., intersections), we decrease the number and spacing of pseudo lanes to avoid overlap and interference as in Fig. 5 (right).

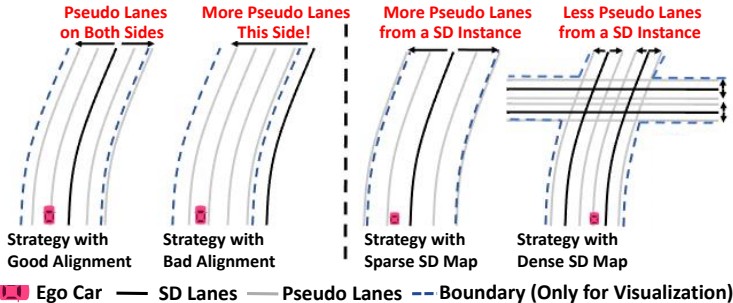

Figure 5: **The Strategy of Adaptive Pseudo Lane Expansion.**

## 4 EXPERIMENTS

### 4.1 DATASETS, METRICS AND BASELINES

**Datasets.** We conduct experiments on the nuScenes dataset (Caesar et al., 2019), which contains 1,000 driving scenes of approximately 20 seconds each. The dataset includes various sensor data such as camera inputs, annotations, and HD maps. While the dataset does not provide SD maps, we follow (Jiang et al., 2024) to extract SD maps from OpenStreetMap (Haklay & Weber, 2008) and align them with the coordinate system of nuScenes. We use trajdata (Ivanovic et al., 2024) interface to access the past and future trajectories of agents. We upsample the trajectories to 10Hz and predict 3 seconds of future trajectories from 2 seconds of past trajectories. Only samples with complete past and future trajectories are used for training and evaluation.

**Metrics.** We follow the metrics in (Gu et al., 2024a;b) as well, with three widely used metrics: minimum Average Displacement Error (minADE), minimum Final Displacement Error (minFDE), and Miss Rate (MR). We predict multiple future trajectories and calculate the ADE and FDE for the trajectory closest to the ground truth, referred to as minADE and minFDE. ADE measures the average $L_2$ distance between predicted and ground-truth trajectories, while FDE measures the $L_2$ distance between the final points of the predicted and ground-truth trajectories. Miss Rate is the proportion of samples where FDE exceeds preset threshold.

**Baselines.** We modified BEVPred (Gu et al., 2024b) as a baseline for comparison under the same protocol. The method inputs BEV features from a pre-trained online mapping model and GT agent history to DenseTNT and HiVT. We additionally provide the model with GT HD maps or SD maps to ensure its input configuration aligns with ours, which is called BEVPred+. We also make a comparison with online mapping based motion prediction methods, which are another approach to reducing the cost of HD map construction. We use original BEVPred and UncentaintyPred (Gu et al., 2024a) as baselines. These methods use HiVT and DenseTNT as base motion prediction model. We implement the them with MTR Shi et al. (2022) ourselves.

### 4.2 RESULTS

**Quantitative Results.** As shown in Table 2, our method achieves significant improvement in performance with both SD map and HD map input. Compared with BEVPred+ which also utilizes raw sensor data, our method further lowers the SD-HD gap, especially on the DenseTNT-based model, which states the effectiveness of our SD map oriented modules.

**SD-HD Gap with Different Base Models.** For base motion prediction models, the anchor-based model DenseTNT is affected more greatly by the precision of the maps than the anchor-free model HiVT. This is due to the different ways these models utilize and depend on maps. Anchor-free model HiVT is designed around agent features, with map features serving as auxiliary information, integrated through cross-attention. This makes the model less dependent on the map, allowing it workers even without any map input. Additionally, the robust map encoder and cross-attention mechanism can still generate useful map features from lower-precision SD maps. In contrast, the anchor-based model DenseTNT directly uses vectorized maps in anchor generation and selection, so the accuracy of the map directly affects the plausibility of goal point distribution and the accuracy of the predicted target. This makes the model highly dependent on map precision, widening the performance gap between HD and SD maps.

Table 2: **Quantitative Results.** BEVpred+ is a baseline that has the same input setting as ours, introduced in Sec. 4.1.

| Method | SD Map Input | | | HD Map Input | | | SD-HD Gap | | |
|---|---|---|---|---|---|---|---|---|---|
| | minADE↓ | minFDE↓ | MR↓ | minADE↓ | minFDE↓ | MR↓ | △ minADE↓ | △ minFDE↓ | △ MR↓ |
| HiVT | 0.3998 | 0.8207 | 0.0918 | 0.3868 | 0.8063 | 0.0870 | 0.0130 | 0.0144 | 0.0048 |
| HiVT + BEVPred+ | 0.3584 | 0.7261 | 0.0702 | 0.3365 | 0.6997 | 0.0683 | 0.0219 | 0.0267 | 0.0019 |
| HiVT + Ours | **0.3128** | **0.6637** | **0.0643** | **0.3092** | **0.6575** | **0.0633** | **0.0036** | **0.0062** | **0.0010** |
| DenseTNT | 1.2117 | 1.9849 | 0.2776 | 0.8809 | 1.4890 | 0.1903 | 0.3308 | 0.4959 | 0.0873 |
| DenseTNT + BEVPred+ | 1.1940 | 2.0029 | 0.3285 | 0.7427 | 1.3419 | 0.1552 | 0.4513 | 0.6610 | 0.1733 |
| DenseTNT + Ours | **0.6854** | **1.2716** | **0.1540** | **0.6612** | **1.1807** | **0.1476** | **0.0242** | **0.0909** | **0.0064** |
| MTR | 0.3732 | 0.7732 | 0.0841 | 0.3464 | 0.7396 | 0.0803 | 0.0268 | 0.0336 | 0.0038 |
| MTR + BEVPred+ | 0.3376 | 0.7132 | 0.0783 | 0.3128 | 0.6892 | 0.0730 | 0.0248 | 0.024 | 0.0053 |
| MTR + Ours | **0.2883** | **0.6359** | **0.0672** | **0.2871** | **0.6340** | **0.0670** | **0.0012** | **0.0019** | **0.0002** |

Table 3: **Comparison with Online HD Mapping Based Methods**

| Base Model | HiVT (Zhou et al., 2022) | | | DenseTNT (Gu et al., 2021) | | | MTR (Shi et al., 2022) | | |
|---|---|---|---|---|---|---|---|---|---|
| Method | minADE↓ | minFDE↓ | MR↓ | minADE↓ | minFDE↓ | MR↓ | minADE↓ | minFDE↓ | MR↓ |
| Base | 0.3657 | 0.7473 | 0.0710 | 0.7664 | 1.3174 | 0.1547 | 0.3504 | 0.7471 | 0.0788 |
| Unc (Gu et al., 2024a) | 0.3588 | 0.7232 | 0.0660 | 0.8123 | 1.3426 | 0.1567 | 0.3189 | 0.7063 | 0.0761 |
| BEVPred (Gu et al., 2024b) | 0.3652 | 0.7323 | 0.0710 | 0.7630 | 1.3609 | 0.1576 | 0.3214 | 0.7105 | 0.0722 |
| Ours | **0.3128** | **0.6637** | **0.0643** | **0.6854** | **1.2716** | **0.1540** | **0.2883** | **0.6359** | **0.0672** |

**Comparison with Online Mapping Based Motion Prediction.** As shown in Tab. 3, our method outperforms the state-of-the-art online mapping based motion prediction approach. Notably, online mapping based methods have the same target as ours so we make the comparison. However, the two approaches use different settings with extra information individually, which is discussed in Sec. 2.2.

**Qualitative Analysis.** We visualize the motion prediction results of our method on anchor-free model in Fig. 6 in three scenarios: driving straight through an intersection, making a right turn at a T-junction, and following a left-curving road. Even when the SD map is not aligned with the road center, our method still accurately predicts future trajectories. Fig. 7 shows the results of the anchor-based model. In the scenario, the vehicle is on the right side of the road and will drive straight. The provided SD instance correctly indicates the road direction but is located far to the left, resulting in candidate target points that are also positioned on the left side, leading to substantial errors. By applying the Pseudo Lane Expansion method, the SD instance is extended laterally, with the parallel line on the right generating candidate target points ahead of the vehicle, allowing the model to accurately select the target point and predict a trajectory closely aligned with the ground truth.

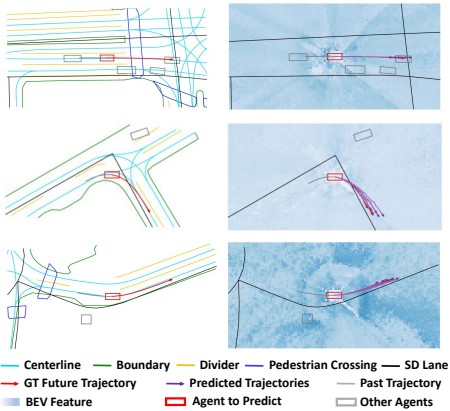

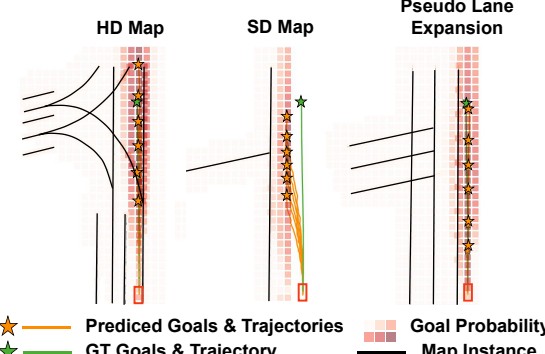

Figure 6: **Qualitative Results of the Anchor-Free Model.** The purple lines with arrows show six predicted future trajectories, and the red lines represent GT future trajectories. The BEV Feature is colored by the max value of the hidden feature at each grid. Our method provides an accurate prediction for future motion.

Figure 7: **Qualitative Results of the Anchor-Based Model.** Left: the detailed HD map generates appropriate anchors in front of the agent. Middle: the sparser and misaligned SD map results in biased anchors. Right: Pseudo Lane Expansion in the model generates extra anchors near the agent and helps predict the right goal and trajectory.

Table 4: **Ablation Study on Pseudo Lane Expansion.**

| Map | Distances | minADE ↓ | minFDE ↓ | MR ↓ |
|---|---|---|---|---|
| HDMap | - | 0.6612 | 1.1807 | 0.1476 |
| SDMap | - | 1.9735 | 3.8357 | 0.5892 |
| SDMap | [0,3] | 0.9627 | 1.5110 | 0.2451 |
| SDMap | [0,3,6] | 0.7941 | 1.3863 | 0.1627 |
| SDMap | [0,2,4] | 0.8472 | 1.3979 | 0.1722 |
| SDMap | [0,3,6,9] | 0.8132 | 1.3855 | 0.1630 |
| SDMap | Adaptive | 0.6854 | 1.2716 | 0.1540 |

Table 6: **Performance of Two Dense Goals Generation Strategy.** PLE refers to Pseudo Lane Expansion and APLE refers to Adaptive Pseudo Lane Expansion.

| Strategy | Kernel Size | minADE ↓ | minFDE ↓ | MR ↓ |
|---|---|---|---|---|
| Original | 2 | 1.9735 | 3.8357 | 0.5892 |
| Original | 6 | 1.2833 | 2.5889 | 0.3451 |
| Original | 10 | 1.3739 | 2.6381 | 0.3890 |
| PLE | 2 | 0.7941 | 1.3863 | 0.1627 |
| APLE | 2 | 0.6854 | 1.2716 | 0.1540 |

Table 5: **Ablation Study on Enhanced Road Observation.** $F_{norm}$ denotes the normalization of head weights. "-" under "Module" means removing the module.

| Module | $F_{norm}$ | minADE ↓ | minFDE ↓ | MR ↓ |
|---|---|---|---|---|
| - | - | 0.3244 | 0.6879 | 0.0832 |
| Enhanced Road Obs. | Softmax | 0.3239 | 0.6918 | 0.0863 |
| Enhanced Road Obs. | Sigmoid | 0.3211 | 0.6879 | 0.0801 |
| Enhanced Road Obs. | None | 0.3128 | 0.6637 | 0.0643 |

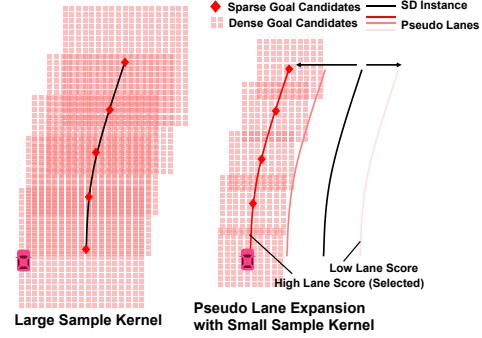

Figure 8: **Large Sample Kernel VS. Pseudo Lane Expansion.**

## 4.3 Ablation Studies

**Enhanced Road Observation.** As Table 5 shows, our Enhanced Road Observation improves the motion prediction performance with effective feature fusion. We test different normalization methods of head weights. Normalizing weights using either Softmax or Sigmoid functions results in poorer performance, while directly using unrestricted weights yields the best performance.

**Hyper-Parameters of Pseudo Lane Expansion.** Table 4 presents the ablation study results for Pseudo Lane Expansion on the DenseTNT-based model. Insufficient or too close expanded lines can limit coverage, potentially missing areas near the vehicle to be predicted, especially when the bias of the SD instance is large. Conversely, an excessive number of lines can lead to interference and increased computational load. Our dynamic strategy adjusts the parameters based on the distribution of SD lanes and achieving the best performance.

**Ablation of Anchor Generation Strategy on DenseTNT.** Our Pseudo Lane Expansion widens the coverage of dense goal points. There is another way to achieve this: using a larger sample kernel to simply generate more dense goal points without modifying sparse ones. However, Pseudo Lane Expansion achieves this in a more efficient and interpretable way because goal point selection is a hierarchical process in DenseTNT. Simply increasing the sampling kernel size to achieve a coverage similar to pseudo-lane expansion would require a kernel size of nearly 10. This would lead to an excessively large number (more than 2000 for a lane) of densely sampled points, significantly reducing model efficiency, which is shown in Fig. 8 (left). In contrast, Pseudo Lane Expansion uses a smaller sample kernel and generates pseudo lanes that hypothesize the approximate locations of potential drivable paths, as shown in Fig. 8 (right). Through lane scoring, the model identifies the pseudo lanes most likely to represent road structures. Dense sampling is then applied only around these selected pseudo-lanes. This approach reduces the number of densely sampled points and is specifically designed to adapt to the coarser and less aligned SD maps. Table 6 shows that our Pseudo Lane Expansion achieves better performance than trivially increasing the size of the sample kernel.

## 5 Conclusion

In this paper, we revisit SD map based motion prediction with end-to-end learning. To address the low resolution and alignment accuracy issues of SD maps, we design two modules called Enhanced Road Observation and Pseudo Lane Expansion. Experiments demonstrate that our method effectively improves the performance of motion prediction and narrows the performance gap between HD and SD maps, which shows the potential of the low-cost SD map based motion prediction paradigm.

## 6 ETHICS STATEMENT

The research conducted in the paper conforms with the ICLR Code of Ethics.

## 7 REPRODUCIBILITY STATEMENT

We describe the proposed module in Sec. 3 and implementation details in Appendix A. The datasets are public accessible. The code and checkpoints will be open-sourced for reproduction.

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

## A  IMPLEMENTATION DETAILS

We train our model on 8 RTX 4090 GPUs with a batch size of 1 on each GPU. We set the learning rate to $2 \times 10^{-4}$ and the number of epochs to 48, with no dropout for faster convergence.

For the BEV encoder, we adopt the official configuration of BEVFormer-base (Li et al., 2022b). The encoder takes a temporal queue of 4 samples as input and obtains BEV features with 6 encoder layers. The BEV feature has a size of 200x200x256 and is in the Lidar coordinate system.

For the motion prediction models, we strictly align the setting of original modules in HiVT and DenseTNT with previous works (Gu et al., 2024a;b) for fair comparison. Specifically, we use a 4-layer temporal transformer, a 1-layer local interaction module, and a 3-layer global interaction module of HiVT. We only add new modules to these two base models without removing existing modules.

For MTR (Shi et al., 2022) model, we integrate BEV-Agent Encoder in Query-Centric Scene-Encoder module and Enhanced Road Observation in each decoder layer. The number of layers and the training hyper-parameters are aligned with default setting.

## B  LLM USAGE

LLMs are used in writing for improving grammar and correcting typos.

