# OpenReview forum: "Revisiting Standard-Definition Map Based Motion Prediction in the Era of End-to-End Autonomous Driving"
_ICLR.cc/2026/Conference — ICLR 2026 Conference Withdrawn Submission_

### Official Review · Reviewer_HunK · 2025-10-29

**Soundness:** 2
**Presentation:** 2
**Contribution:** 3
**Rating:** 4
**Confidence:** 3

**Summary:**

This paper focuses on improving motion prediction using SD maps. The authors propose two novel modules: Enhanced Road Observation and Pseudo Lane Expansion. The core contribution is narrowing the SD-HD performance gap, achieving performance comparable to HD maps with SD maps. The proposed modules claim to significantly enhance prediction accuracy.

**Strengths:**

1. The presentation of this paper is good. The figures are well-constructed and effectively communicate the core ideas and implementation details of the proposed method.
2. The proposed method could be described as demonstrating originality and, to a certain extent, bridging the performance gap between HD and SD maps, thus addressing limitations in prior research.

**Weaknesses:**

1. MTR is an anchor-based method, and the authors' emphasis on anchor-free in the introduction indicates a misunderstanding of the related work.
2. Is the Adaptive Pseudo Lane Expansion mechanism reasonable? The mechanism is overly simplistic, and the so-called "adaptive" approach resembles a rule-based method. While it is intriguing, there is still room for improvement. Additionally, the name of this module is inappropriate.
3. The qualitative results of the anchor-free and anchor-based models should include a comparison between the proposed method and baseline methods to demonstrate the effectiveness of the innovations introduced in this paper.
4. The experimental comparisons should appropriately integrate more recent trajectory prediction methods (e.g., methods developed after 2024).

**Questions:**

1. Is the authors' approach of "expanding reference points by creating parallel lines" overly idealistic? In cases where there are non-parallel road structures in certain urban environments, is there a mechanism to address this?
2. Are the two innovations mentioned by the authors based on a common idea? Considering the overall approach, does this innovation seem somewhat insubstantial?
3. What is the impact of the proposed method on the model's inference speed?

---

### Official Review · Reviewer_55XX · 2025-10-30

**Soundness:** 3
**Presentation:** 2
**Contribution:** 2
**Rating:** 4
**Confidence:** 4

**Summary:**

This paper revisits the use of Standard-Definition (SD) maps for motion prediction in the era of end-to-end autonomous driving. Unlike traditional High-Definition (HD) maps, SD maps are low-cost but suffer from low resolution and poor alignment. The authors propose two novel modules to compensate for the limitations of SD maps. ERO enhances feature fusion between BEV and SD map features by expanding deformable attention reference points along parallel lines, while PLE improves anchor generation in anchor-based models by creating pseudo lane candidates around SD map instances. Extensive experiments on nuScenes with multiple base models (HiVT, DenseTNT, MTR) demonstrate that the proposed approach significantly narrows the performance gap between SD and HD maps, achieving comparable accuracy while greatly reducing map dependency and cost.

**Strengths:**

1、The paper addresses the high annotation cost of HD maps with a practical and cost-effective alternative using SD maps, aligning well with current trends in autonomous driving.
2、The authors clearly identify the key limitations of SD maps (low resolution and misalignment) and introduce two targeted modules to mitigate these issues.
3、The method is validated across multiple representative architectures (HiVT, DenseTNT, and MTR), showing consistent and significant improvements, which demonstrates the robustness and effectiveness of the proposed approach.

**Weaknesses:**

1、The proposed modules (ERO and PLE) are extensions of existing mechanisms such as deformable attention and anchor sampling, making the contribution more of an engineering optimization than a theoretical innovation.
2、The ERO module fundamentally compromises on the limitations of SD maps. By heuristically expanding sampling points along coarse SD map directions, it effectively performs a blind search around the map to find useful BEV features, rather than addressing the root cause，the lack of precise lane-level geometry. Its performance heavily depends on the BEV encoder’s ability to extract reliable lane semantics, which limits robustness.
3、The PLE module lacks interpretability and theoretical justification. Its adaptive pseudo-lane generation strategy relies on heuristic rules that are dataset-specific and empirically tuned, raising concerns about generalization to diverse cities or driving environments.
4、The proposed components are tightly coupled with specific architectures (ERO with deformable attention, PLE with anchor-based models), limiting their generality as a universal SD map compensation strategy.

**Questions:**

The paper tackles a practically meaningful problem，reducing the dependency of motion prediction on costly HD maps. However, its contributions are mainly heuristic and incremental rather than conceptual or methodological. The two proposed modules are specifically tailored for different existing frameworks instead of providing a unified or generalizable solution. Moreover, the experiments lack sufficient validation of generalization across datasets, environments, and map sources. Overall, while the work has engineering value, it falls short of the novelty and rigor expected for top-tier venues.

---

### Official Review · Reviewer_CEdP · 2025-10-31

**Soundness:** 3
**Presentation:** 3
**Contribution:** 3
**Rating:** 6
**Confidence:** 2

**Summary:**

This paper revisits the problem of motion prediction for autonomous driving, with an emphasis on using low-cost, standard-definition (SD) maps accessible from services like OpenStreetMap, in place of expensive, labor-intensive high-definition (HD) maps. The authors extend both anchor-based and anchor-free end-to-end motion prediction models to fuse raw sensor data with SD map inputs. Two new modules are introduced: Enhanced Road Observation (for improved SD map-sensor feature fusion in anchor-free models) and Pseudo Lane Expansion (for robust anchor generation in anchor-based models when SD maps are coarse or misaligned). Experiments on the nuScenes benchmark, with SD maps aligned from OSM, demonstrate that the SD-HD performance gap can be narrowed substantially, with the proposed method sometimes outperforming state-of-the-art online HD map-based approaches. Ablations and qualitative illustrative figures further support the contributions.

**Strengths:**

1. The cost and lack of scalability for HD map production in autonomous driving is a real industrial bottleneck. Revisiting SD map-based motion prediction, leveraging contemporary end-to-end models, addresses a practical and underexplored aspect in the field.
2.  Extensive experiments on nuScenes with SD maps (Tables 2, 3), comparisons to recent online HD mapping approaches, and detailed ablation studies (Tables 4–6, Figures 6–8) systematically validate both the necessity and value of the main modules.
3. Most methodological descriptions, including equations, algorithmic notations (Sec. 3.2–3.4), and architecture diagrams, are detailed and precise.  Sectional flow is logical—from motivation to problem decomposition, method specification, and in-depth analysis.

**Weaknesses:**

1. While some alternatives are explored (Tables 4–6), the analysis of Enhanced Road Observation and Pseudo Lane Expansion does not sufficiently benchmark against other plausible strategies such as more sophisticated probabilistic map alignment, soft-assignment for anchor generation, or hybrid attention mechanisms. For example, Pseudo Lane Expansion is compared mainly to varying kernel size (Table 6, Figure 8), but there is no direct experimental comparison to data-driven anchor point proposals (as in VectorNet or interactive GNN approaches).
2. The main evaluation is on nuScenes with OSM-based SD maps.   There is little discussion or experimental validation on how well these findings generalize across cities, map providers, or regions with lower OSM quality.   It isn’t clear how robust the approach is when SD map misalignment errors are systematically higher, or whether the modules could be used as-is with different agent types or in multi-agent scenarios (e.g., non-vehicle classes).

**Questions:**

1. How robust are the proposed methods in environments where SD map quality is extremely poor or in urban areas with dense, overlapping roads?  Have you tested on cities/regions where OSM is less reliable or aligns less well than in nuScenes?  Any quantitative metrics on SD map misalignment sensitivity?
2. Does the parallel line procedure always generate plausible reference points, and can this approach be degenerate in multi-lane intersections?
3. Can you showcase more systematically where the new modules fail or provide less benefit—either quantitatively (e.g., boxplots across scenarios) or visually (e.g., particularly difficult cases for adaptive pseudo-lane expansion)?

---

### Official Review · Reviewer_KF8p · 2025-11-01

**Soundness:** 2
**Presentation:** 3
**Contribution:** 2
**Rating:** 4
**Confidence:** 4

**Summary:**

This paper revisits the use of low-cost standard-definition (SD) maps for motion prediction, aiming to close the significant performance gap compared to expensive high-definition (HD) maps in the context of end-to-end autonomous driving. The authors first demonstrate that integrating raw sensor (image) features within an end-to-end framework substantially narrows this performance gap. They further identify and analyze the unique challenges that coarse and misaligned SD maps introduce, specifically for feature fusion in anchor-free models and anchor generation in anchor-based models. To overcome these issues, the paper proposes two novel modules: enhanced road observation to improve feature fusion and pseudo lane expansion to create better anchor distributions.

**Strengths:**

1. The paper addresses a practical problem in autonomous driving of the high cost and scalability issues of HD maps. The motivation to leverage low-cost, widely available SD maps as a viable alternative is well-argued.
2. The proposed method achieves state-of-the-art performance by a large gap.

**Weaknesses:**

1. The paper's cost-efficiency argument is debatable. While it avoids costly HD map annotations for training, it introduces a strict dependency on aligned SD map GT at inference time. This reliance on an external map feed contrasts with online mapping approaches (e.g., BEVPred) which, once trained, generate map elements internally and can operate without any external map GT, potentially making them more self-contained at deployment.
2. While the method is designed to handle misalignment and low resolution, its robustness to topologically outdated SD maps is not evaluated. Real-world scenarios like new road constructions, changed intersections, or permanent road closures are not analyzed.
3. The qualitative analysis is sparse (Figs. 6 & 7) and primarily serves as a positive illustration of success cases. The paper would be much stronger with a more in-depth analysis of challenging scenarios or failure modes to provide deeper insights beyond the quantitative metrics.
4. The paper omits an analysis of its limitations. This leaves potential failure modes, the computational overhead of the new modules, and the precise boundaries of the method's applicability unaddressed.

**Questions:**

The paper presents an intuitive approach to leveraging SD maps to mitigate the high costs of HD maps. However, this raises questions about its practical efficiency and real-world applicability. A key motivation is cost-saving by avoiding HD map training data, but the proposed method may require more cost by leveraging SD map GT at inference time. The authors should elaborate on the trade-offs of this inference-time dependency, especially compared to online mapping methods like BEVPred, which can operate without external map GT after being trained. Furthermore, the model targets autonomous driving, but its validation in real-world scenarios is insufficient. Analysis is needed on its performance when using noisy inputs from an actual perception system (instead of GT agent information) and its robustness to topologically outdated maps (e.g., construction, road changes), not just map misalignment.

---

### Note · Authors · 2025-11-13

I have read and agree with the venue's withdrawal policy on behalf of myself and my co-authors.